# Gender with marital status, cultural differences, and vulnerability to hypertension: Findings from the national survey for noncommunicable disease risk factors and mental health using WHO STEPS in Bhutan

Hiromi Kohori Segawa[1,2], Hironori Uematsu[1], Nidup Dorji[3], Ugyen Wangdi[3], Chencho Dorjee[3], Pemba Yangchen[4], Susumu Kunisawa[1], Ryota Sakamoto[5], Yuichi Imanaka[1]*

1 Department of Healthcare Economics and Quality Management, Graduate School of Medicine, Kyoto University, Kyoto City, Kyoto, Japan, 2 Kokoro Research Center, Kyoto University, Kyoto City, Kyoto, Japan, 3 Faculty of Nursing and Public Health, Khesar Gyalpo University of Medical Sciences of Bhutan, Thimphu, Kingdom of Bhutan, 4 Department of Non-Communicable Disease, Ministry of Health in Bhutan, Thimphu, Kingdom of Bhutan, 5 Centre for Southeast Asian Studies, Kyoto University, Kyoto City, Kyoto, Japan

* imanaka-y@umin.net

**Data Availability Statement:** Since the Ministry of Health, Bhutan is the authority for the data, the

## Abstract

Cardiovascular disease is the leading cause of death in the Kingdom of Bhutan. Thus, the early detection and prevention of hypertension is critical for reducing cardiovascular disease. However, the influence of sociocultural factors on vulnerability to hypertension needs further investigation. This study performed secondary data analysis on 1,909 individuals in a cross-sectional study (the National survey for noncommunicable disease risk factors and mental health using World Health Organization (WHO) STEPS approach in Bhutan– 2014). Multivariate logistic regression demonstrated a significant association between gender with marital status and hypertension. Women had a higher odds ratio than men (Ref) when married (AOR: 1.27, 95% confidence intervals (CI): 1.23–1.31), and when separated, divorced, or widowed (AOR: 1.18, 95% CI: 1.12–1.26). People who speak the Tshanglakha language scored the highest odds (AOR: 1.24, 95% CI: 1.20–1.27), followed by Lhotshamkha (AOR: 1.09, 95% CI: 1.06–1.12) and Dzongkha (Ref) after adjusting for various social and biomedical factors. Additionally, tobacco use displayed decreased odds for hypertension. To promote the early detection and prevention of hypertension, these cultural factors should be considered even within small geographic areas, such as Bhutan. It is necessary to strengthen hypertension preventive strategies for people who speak Tshanglakha and Lhotshamkha. Furthermore, careful consideration should be given to preventing hypertension among adults aged 40 years or more, women who are married, separated, divorced, or widowed, and men who never married in Bhutan.

authors were not able to make it available with this manuscript. However, the original report (World Health Organization and Ministry of Health in Bhutan. National survey for noncommunicable disease risk factors and mental health using WHO STEPS approach in Bhutan—2014) is available from https://apps.who.int/iris/handle/10665/204659, ISBN 9789290225003. Requests to access the raw data can be directed to the Research Ethics Board of Health at the Royal Government of Bhutan or the Ministry of Health, Bhutan. Contact: P.O. Box: 726, Kawajangsa, Thimphu, Bhutan; Phone No: +975-2-328095, 321842, 322602, 328091.

**Funding:** This study was supported by 1) Hiromi Kohori-Segawa, K. MATSUSHITA FOUNDATION, Grant Numbers 19-048 http://matsushita-konosuke-zaidan.or.jp/en/ 2) Yuichi Imanaka, Program of Collaborative Research at the Centre for Southeast Asian Studies (IPCR-CSEAS), Kyoto University 2018 type4 https://ipcr.cseas.kyoto-u.ac.jp/ 3) Hiromi Kohori-Segawa, Program of Collaborative Research at the Centre for Southeast Asian Studies (IPCR-CSEAS), Kyoto University 2020 type7 https://ipcr.cseas.kyoto-u.ac.jp/ 4) Yuichi Imanaka, JSPS KAKENHI Grant Numbers JP19H01075 from the Japan Society for the Promotion of Science https://www.jsps.go.jp/english/ The funders played no role in the study design, data collection and analysis, decision to publish, or preparation of the manuscript.

**Competing interests:** The authors declare no potential financial conflict of interest related to the manuscript. The funders played no role in the study design, data collection and analysis, decision to publish, or preparation of the manuscript.

## Introduction

According to the World Health Organization (WHO), 38 million people die from noncommunicable diseases (NCDs) annually. Approximately three-quarters of these deaths (28 million) occur in low- and middle-income countries [1]. The increase in NCDs has led to an economic burden on individuals, families, society, and the global community [2]. Hypertension is the most critical global risk factor for morbidity and mortality, being one of the strongest risk factors for nearly all forms of cardiovascular disease acquired during one's lifetime. In addition, asymptomatic changes in several organs associated with hypertension are a vital indicator of the continued progression of cardiovascular disease [3]. Strategies such as screening, early detection, counseling, and continued follow-up of people with hypertension are required to prevent cardiovascular-related diseases [4]. At the same time, nonmodifiable factors, such as age, sex, menopause, and genetics can predispose individuals to hypertension [5]. Several studies have reported that hypertension is significantly related to chronic stress, race, place of residence, and social factors [4–9].

Similarly, in the Kingdom of Bhutan (hereafter referred to as Bhutan), mortality due to NCDs has increased from 53% (2008) to 69% (2016) [10, 11]. In particular, cardiovascular disease was the leading cause of mortality in 2016 [11]. Therefore, the Royal Government of Bhutan began developing measures against NCDs. Studies in Bhutan have concluded that primary prevention strategies, such as eating a healthy diet, avoiding alcohol and tobacco, and regular exercise, are necessary to prevent NCDs [12–14]. The results warrant further verification to promote practical knowledge and skills regarding disease prevention [15]. Furthermore, people in Bhutan want to remain healthy and happy, and they perceived that health is a major factor for happiness. However, people were not aware of how to protect their health in their daily lives [15]. Concerningly, some of the participants had never had their blood pressure taken despite having hypertension. Moreover, some were unaware why they needed to have their blood pressure assessed periodically [15]. Therefore, further prevention strategies should be developed, in addition to the promotion of primary prevention, with considerable attention given to vulnerable people in Bhutan. However, to the best of our knowledge, there have been few studies in Bhutan investigating issues related to preventing hypertension, such as examining social and cultural factors. Specifically, investigations using objective variables (such as blood pressure measurement) in surveys or biomarkers are scarce in Bhutan.

Thus, the current study focused on the prevention of hypertension in Bhutan, especially relating to identifying the sociocultural factors correlated with hypertension. The findings will contribute to the field and lead to developing measures for the prevention of hypertension.

### Objectives

This study aimed to investigate the sociocultural factors associated with hypertension in Bhutan.

## Materials and methods

### Study design

A cross-sectional study was conducted using data from the "National survey for noncommunicable disease risk factors and mental health using WHO STEPS approach in Bhutan– 2014" [12].

### Study setting

Bhutan is located in the eastern Himalayas between India in the east, west, and south, and China in the north. It spans 38,394 km$^2$ with a population of 748,931 in 2020. The estimated

gross domestic product per capita was $3,411.94 (US) in 2019 [16]. Bhutan bases its national development strategy on the philosophy of gross national happiness, which aspires for sustainable development and happiness for all citizens [17, 18].

## Data source

Data were derived from the "National survey for noncommunicable disease risk factors and mental health using WHO STEPS approach in Bhutan—2014" conducted by the WHO and the government of Bhutan [12]. The target sample size was 2,912, which was considered sufficient to represent the target population (adults aged 18–69 years) in Bhutan (S 1). Multistage cluster sampling combined with probability proportional to size and systematic random sampling were used to ensure a representative sample. The urbanicity strata level was considered (rural:urban = 7:3); an area block was designated as the cluster level (n = 182), selected from gewogs (group of villages). The Kish method was applied to select participants from each household using age and gender as variables and the sampling framework from the "Population and Housing Census of Bhutan 2005" [19–21]. Trained staff collected data through face-to-face interviews. There were 2,822 valid respondents, a response rate of 97% (S 1). Details of the survey procedure have been previously reported [12], and we quoted some additional points from the "National survey for noncommunicable disease risk factors and mental health using WHO STEPS approach in Bhutan– 2014" [12] (S 1).

## Definition of variable

**Dependent variable.** Hypertension was defined by the following criteria: (1) systolic blood pressure (SBP) $\geq$ 140 mmHg or diastolic blood pressure (DBP) $\geq$ 90 mmHg, measured as the average of three measurements taken by health staff in the survey; (2) a previous diagnosis of hypertension by healthcare workers; (3) currently taking medication for hypertension [19].

**Explanatory variables.** Explanatory variables (i.e., sociocultural, lifestyle, biological, and mental health) were selected and categorized following the WHO guidelines [19] and previous literature [22–35]. Sociocultural variables included gender and marital status (hereafter gender-marital status), which we combined due to an interaction effect on hypertension (S1 and S2 Tables, S1–S3 Figs) (men-married or cohabitant, men never-married, men-separated or divorced or widowed, women-married or cohabitant, women never-married, women-separated or divorced or widowed). Furthermore, age, education, work status, place of residence (urbanicity), income, and language were also considered. Level of education was categorized as no formal education, elementary (1–10 years of education), high school (11–12 years of education), or tertiary (more than 12 years of education); a promotion test is given in the 10th and 12th years in Bhutan. Work status was defined as employed, self-employed, or nonworking. Residential area was considered as either urban or rural. Income was allocated into quartiles. Survey languages included Dzongkha, Tshanglakha, Lhotshamkha, and English.

Lifestyle variables included tobacco use, alcohol consumption, and fruit and vegetable intake. Tobacco use was rated as never, currently using tobacco, or any prior use of tobacco (not only smoking but also chewing tobacco) [19, 35]. Alcohol consumption was categorized as never, a light or moderate drinker, or a heavy drinker [19, 29]. The consumption of fruits and vegetables was rated as less than five serves per day or five or more serves per day. One serve of fruit was defined as a medium-sized banana or apple or equivalent, and one serve of vegetables was equal to one cup of green leafy vegetables or half a cup of cooked vegetables [19]. Other lifestyle factors included physical activity ($\geq$150 min of moderate activity per week and less than 150 min of moderate activity per week) [19] and estimated salt intake (less than or more than 5 g per day: this cut-off point was determined according to the WHO guidelines).

The Tanaka formula was used to estimate salt intake [22, 30–34]: Salt Intake per day(g) = (21.98 * ((Spot Urinary Sodium(mmol/l)/(Spot Urinary Creatinine(mg/dl) *10/0.0884)) * (((14.89* Weight (Kg))+(16.14*height(m))*(2.04*age))-2244.45))**0.392)/17.1).

Biomedical and mental health variables included blood glucose, total cholesterol, body mass index, family history, cardiovascular disease, and suicide ideation. Blood glucose was categorized as normal (fasting blood glucose <110 mg/dl), abnormal (fasting blood glucose ≥110 mg/dl, under medication or previously diagnosed with diabetes) [19]. Total cholesterol was categorized as <240 mg/dl, ≥240 mg/dl, or under medication [19]. Levels of BMI included <18.5, 18.5–24.9, 25–29.9, and ≥30 kg/m$^2$ [19]. Family history was assessed as either yes (positive) or no (negative) to the question, "Have any of your blood family members been diagnosed with the following diseases: elevated blood pressure?" The incidence of cardiovascular disease (yes/positive, no/negative) was assessed by asking, "Have you ever had a heart attack or chest pain from heart disease or a stroke?" Suicide ideation was assessed as yes (positive) or no (negative) by asking, "During the past 12 months, have you seriously considered attempting suicide?" All variables were measured using the WHO guidelines.

**Statistical analysis.** Descriptive statistics were reported to describe the characteristics of the study sample. Bivariate logistic regression analyses were conducted to evaluate the associations between the dependent variable and each explanatory variable. Lastly, multiple logistic regression analyses were performed to adjust for all explanatory variables.

We performed complete case analysis. In other words, participants with missing data were excluded from the analysis. In addition, pregnant women were excluded due to their differences in behavior and biological characteristics. All estimates were presented with 95% confidence intervals. The analysis was adjusted for sampling weight to ensure representativeness; these data were also generated from the STEPS Survey. Sampling weight was back-calculated using the demographic proportions of sample units by gender, age, and cluster, and the weighting for each sample from the original 2,822 respondents was further calculated. Data were analyzed using IBM SPSS Statistics version 23 (IBM Corp., Armonk, NY, USA).

In addition, subgroup and sensitivity analyses were conducted. Subgroup analyses were performed with people over 40 years of age, considering the effects of aging and hormonal changes noted in previous studies [36, 37]. Hypertension in subgroup analyses was defined by the following criteria: (1) systolic blood pressure (SBP) ≥ 140 mmHg or diastolic blood pressure (DBP) ≥ 90 mmHg, measured as the average of three measurements taken by health staff in the survey; (2) a previous diagnosis of hypertension by healthcare workers; (3) currently taking medication for hypertension [19].

Sensitivity analyses were conducted to consider differences in vulnerability by hypertension severity. Alternate definitions of hypertension according to severity category were analyzed, including mean SBP ≥180 mmHg or DBP ≥120 mmHg and SBP ≥160 mmHg or DBP ≥100 mmHg [19].

**Research ethics.** The Research Ethics Boards of Health, Ministry of Health of the Royal Government of Bhutan (No. REBH/Approval/2018/089), and the Ethics Committee of the Kyoto University Hospital and Graduate School of Medicine (No. R1796) approved the study protocol. The above ethical committees waived the requirement for individual informed consent as this secondary analysis only used de-identified data. We have submitted a written pledge of confidentiality to the Ministry of Health of the Royal Government of Bhutan.

## Results

There were 1,909 participants with complete data (Fig 1). Table 1 displays the sociodemographic characteristics of all participants, including both non-weighted and weighted data.

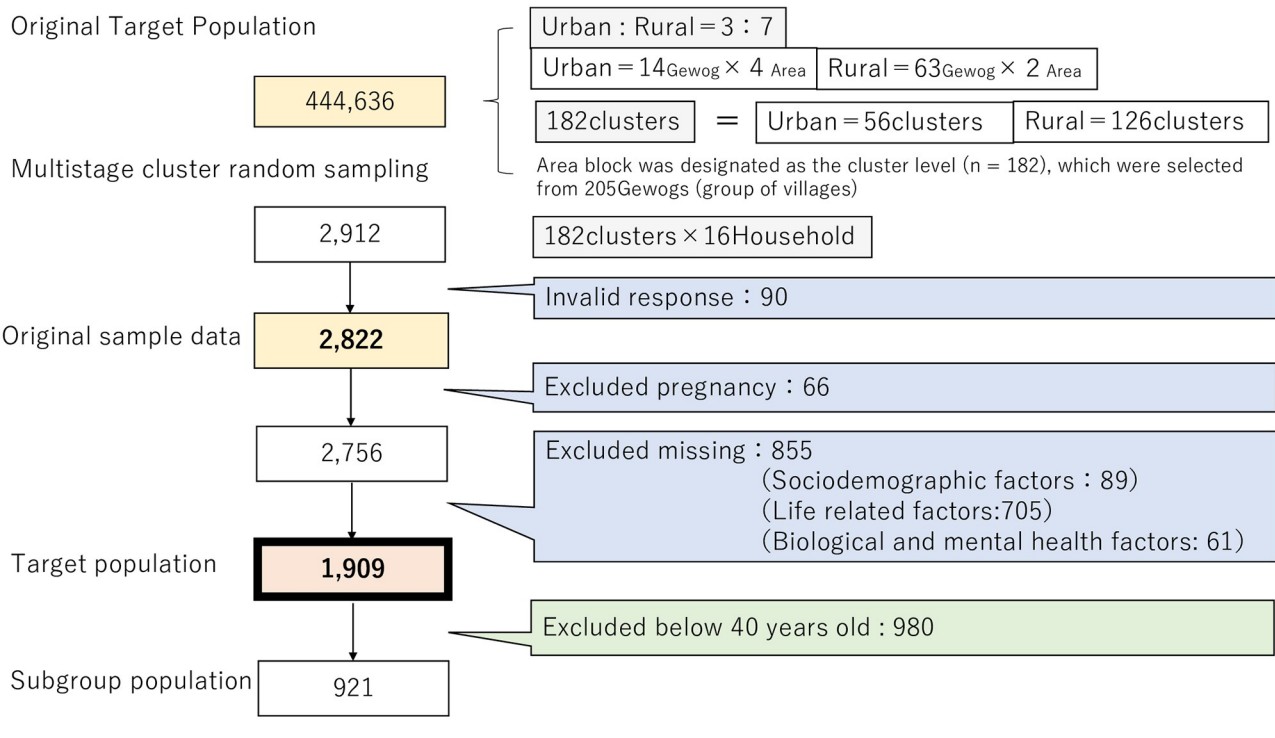

**Fig 1. Target population.**

While the trend remained the same between the non-weighted and weighted data, a difference in statistical power was observed.

The participants were distributed as follows: men (746, 39.1%) and women (1163, 60.9%) without weight adjustment. The majority were married or cohabitating (80.9%). The mean age was 40.2 years (SD: 12.3). More than half of the participants (50.7%) had no formal education, and 48.7% were self-employed. Lastly, 45% were diagnosed with hypertension. Table 2 displays the results of bivariable logistic regression analysis for hypertension, and Table 3 presents the results of multivariable logistic regression analysis for hypertension.

In terms of the sociocultural variables, hypertension was significantly associated with age (30–39 years [AOR: 2.14; 95% CI: 1.56–2.94], 40–49 years [AOR: 3.60; 95% CI: 2.57–5.04], 50–59 years [AOR: 4.78; 95% CI: 3.27–7.00], and 60–69 years [OR: 7.50; 95% CI: 4.75–11.85]) and survey language (Tshanglakha [AOR: 1.72, 95% CI: 1.31–2.25], Lhotshamkha [AOR: 1.37, 95% CI: 1.05–1.79], and English [AOR: 0.31; 95% CI: 0.10–1.00]). Level of education, working status, income level, and residential area were not significantly related to hypertension.

Regarding gender with marital status, never-married men (AOR: 2.22; 95% CI: 1.23–4.01) displayed increased odds of acquiring hypertension compared to married or cohabitant men [Reference: AOR:1]. Multivariable analysis of the weighted data revealed that married or cohabiting women (AOR: 1.27; 95% CI: 1.23–1.31) or separated, divorced, or widowed women (AOR: 1.18; 95% CI: 1.12–1.25) exhibited increased odds of having hypertension compared with married men [Reference: AOR:1] or never-married women (AOR: 0.44; 95% CI: 0.40–0.49). Moreover, separated, divorced, or widowed men (AOR: 0.66; 95% CI: 0.60–0.71) displayed lower odds of hypertension compared to separated, divorced, or widowed women (AOR: 1.18; 95% CI: 1.12–1.25).

In terms of lifestyle variables (Table 3), light or moderate drinkers (AOR: 1.27; 95% CI: 1.00–1.62) and heavy drinkers (AOR: 1.55; 95% CI: 1.18–2.04) had increased odds of

**Table 1. Distribution of sociodemographic variables.**

| | | Total(n) | Hypertension [1] | | |
|---|---|---|---|---|---|
| | | | Non-weighted(n) | Non-weighted | Weighted |
| Total | | 1909 | 1268 | 45.0% | 40.9% |
| Gender-Marital Status | Men-Married or cohabitant | 630 | 286 | 45.4% | 41.2% |
| | Men-Never married | 77 | 27 | 35.1% | 32.0% |
| | Men-Separated or Divorced or Widowed | 39 | 19 | 48.7% | 44.4% |
| | Women-Married or cohabitant | 915 | 424 | 46.3% | 45.2% |
| | Women-Never married | 71 | 10 | 14.1% | 10.5% |
| | Women-Separated or Divorced or Widowed | 177 | 101 | 57.1% | 54.9% |
| Age | 18–29 years | 428 | 92 | 21.5% | 20.7% |
| | 30–39 years | 560 | 235 | 42.0% | 41.6% |
| | 40–49 years | 464 | 255 | 55.0% | 52.8% |
| | 50–59 years | 297 | 179 | 60.3% | 62.1% |
| | 60–69 years | 160 | 106 | 66.3% | 68.8% |
| Education-years | No formal education | 1175 | 596 | 50.7% | 47.3% |
| | 1–10 years | 583 | 234 | 40.1% | 37.5% |
| | 11–12 years | 96 | 16 | 16.7% | 18.4% |
| | More than 12 years | 55 | 21 | 38.2% | 38.9% |
| Working Status | Employee | 328 | 134 | 40.9% | 38.6% |
| | Self-employed | 1059 | 516 | 48.7% | 44.6% |
| | Non-working | 522 | 217 | 41.6% | 38.1% |
| Residential area | Rural | 1320 | 613 | 46.4% | 42.0% |
| | Urban | 589 | 254 | 43.1% | 41.0% |
| Income | Nu.0-9,000 | 707 | 327 | 46.3% | 44.6% |
| | Nu.9,001–30,000 | 741 | 357 | 48.2% | 42.4% |
| | Nu.30,001–60,000 | 555 | 230 | 41.4% | 33.8% |
| | Nu.60,001- | 732 | 316 | 43.2% | 42.0% |
| Survey language | Dzongkha | 679 | 269 | 39.6% | 40.4% |
| | Tshanglakha | 667 | 362 | 54.3% | 47.8% |
| | Lhotshamkha | 531 | 232 | 43.7% | 39.2% |
| | English | 32 | 4 | 12.5% | 12.0% |
| Tobacco use | Never use | 1518 | 713 | 47.0% | 43.3% |
| | Currently use | 391 | 154 | 39.4% | 36.9% |
| Alcohol consumption | Never drink | 962 | 388 | 40.3% | 37.6% |
| | Light or moderate drinking | 536 | 252 | 47.0% | 43.5% |
| | Heavy drinking | 411 | 227 | 55.2% | 47.9% |
| Fruit and vegetable consumption | More than 5 serves per day | 648 | 316 | 48.8% | 44.0% |
| | Five or fewer serves per day | 1261 | 551 | 43.7% | 40.5% |
| Physical Activity | 150 min or more per week | 1760 | 805 | 45.7% | 41.9% |
| | Less than 150 min per week | 149 | 62 | 41.6% | 38.8% |
| Salt intake | Below 5g per day | 21 | 9 | 42.9% | 38.9% |
| | Above or 5g per day | 1888 | 858 | 45.4% | 41.7% |
| Blood glucose | Normal | 1758 | 761 | 43.3% | 39.8% |
| | Abnormal | 151 | 106 | 70.2% | 65.1% |
| Total cholesterol | Lower than 240 mg/dl | 1870 | 843 | 45.1% | 41.3% |
| | 240 mg/dl or higher | 39 | 24 | 61.5% | 60.9% |

(*Continued*)

**Table 1.** (Continued)

| | | Total(n) | Hypertension [1] | | |
|---|---|---|---|---|---|
| | | | Non-weighted(n) | Non-weighted | Weighted |
| BMI | <18.5 | 67 | 19 | 28.4% | 23.1% |
| | $18.5 \leq$ BMI $< 25.0$ | 1118 | 435 | 38.9% | 36.4% |
| | $25.0 \leq$ BMI $< 30.0$ | 568 | 311 | 54.8% | 51.6% |
| | $\geq 30.0$ | 156 | 102 | 65.4% | 65.2% |
| Family history of hypertension | Negative | 1277 | 561 | 43.9% | 41.1% |
| | Positive | 632 | 306 | 48.4% | 42.8% |
| Cardiovascular disease | Negative | 1893 | 859 | 45.4% | 41.7% |
| | Positive | 16 | 8 | 50.0% | 42.7% |
| Considering suicide | Negative | 1867 | 853 | 45.7% | 41.7% |
| | Positive | 42 | 14 | 33.3% | 38.8% |

[1] Hypertension was defined by the following criteria: (1) systolic blood pressure (SBP) $\geq$ 140 mmHg or diastolic blood pressure (DBP) $\geq$ 90 mmHg, measured as the average of three measurements taken by health staff in the survey; (2) a previous diagnosis of hypertension by healthcare workers; (3) currently taking medication for hypertension [19].

hypertension. There was no significant association with salt intake in the non-weighted data; however, multivariable analysis using weighted data showed that more than 5 g of salt intake per day (AOR: 1.41; 95% CI: 1.24–1.60) displayed increased odds of having hypertension. In contrast, current tobacco use (AOR: 0.75; 95% CI: 0.57–0.98) was associated with lower odds of having hypertension. Furthermore, hypertension was not significantly associated with fruit and vegetable consumption or physical activity in the multivariable analysis of the non-weighted data. Multivariable analysis of the weighted data revealed that consumption of five serves of fruit and vegetables or fewer per day (AOR: 0.96; 95% CI: 0.94–0.99) and less than 150 min of physical activity per week (AOR: 0.82; 95% CI: 0.78–0.86) decreased the odds of having hypertension.

Table 4 displays the results of subgroup analysis (only for adults 40 years of age or more). The correlation coefficient between marital status and age was not too high (r = 0.34). Married or cohabitating women (AOR: 1.58; 95% CI: 1.11–2.26) and separated, divorced, or widowed women (AOR: 1.71; 95% CI: 1.07–2.73) displayed increased odds of hypertension compared with men. The subgroup analysis with weighted data displayed similar trends to the multivariable analysis using weighted data, excluding fruit and vegetable consumption.

In contrast to the whole sample, multivariable analysis of the weighted data from adults aged 40 years or more (Table 4) revealed five servings of fruit and vegetables or less per day (AOR: 1.21; 95% CI: 1.16–1.27) increased the odds of having hypertension.

Regarding the biomedical variables (Table 3), abnormal blood glucose (AOR: 2.13; 95% CI: 1.43–3.17), BMI (25.0–29.9 kg/m$^2$ [AOR: 2.61; 95% CI: 1.42–4.78] and $\geq$30.0 kg/m$^2$ [AOR: 3.64; 95% CI: 1.84–7.21]), and a positive family history (AOR: 1.35; 95% CI: 1.09–1.69) increased the odds of having hypertension. Total cholesterol $\geq$240 mg/dl (AOR: 1.66; 95% CI: 1.52–1.81) displayed increased odds of having hypertension in the multivariable analysis of the weighted data. In contrast, cardiovascular disease (AOR: 0.65; 95% CI: 0.56–0.76) exhibited decreased odds of having hypertension in the multivariable analysis of the weighted data.

There was no significant association between mental health variables and having hypertension in the multivariable analysis of the non-weighted data (Table 3). However, the multivariable analysis using the weighted data revealed increased odds of having hypertension with positive suicide ideation (AOR: 1.24; 95% CI: 1.14–1.35).

**Table 2. Bivariable logistic regression analysis for hypertension with sociocultural, lifestyle, biomedical, and mental health variables (n = 1,909).**

| n = 1909 | | Non-weighted | | | Weighted | | |
|---|---|---|---|---|---|---|---|
| | | OR | 95%CI | p-value | AOR[1] | 95%CI | p-value |
| Gender-Marital Status | Men-Married or cohabitant | Ref | (-) | | Ref | (-) | |
| | Men-Never married | 0.54 | (0.35–0.83) | 0.005 | 0.53 | (0.51–0.55) | <0.001 |
| | Men-Separated or Divorced or Widowed | 1.02 | (0.60–1.74) | 0.942 | 0.92 | (0.87–0.99) | 0.017 |
| | Women-Married or cohabitant | 1.01 | (0.85–1.21) | 0.872 | 1.10 | (1.08–1.12) | <0.001 |
| | Women-Never married | 0.29 | (0.18–0.47) | <0.001 | 0.28 | (0.26–0.30) | <0.001 |
| | Women-Separated or Divorced or Widowed | 1.56 | (1.17–2.08) | 0.003 | 1.75 | (1.68–1.83) | <0.001 |
| Age | 18–29 years | Ref | (-) | | Ref | (-) | |
| | 30–39 years | 2.23 | (1.76–2.83) | <0.001 | 2.26 | (2.21–2.32) | <0.001 |
| | 40–49 years | 4.25 | (3.32–5.43) | <0.001 | 4.06 | (3.94–4.18) | <0.001 |
| | 50–59 years | 4.67 | (3.56–6.11) | <0.001 | 5.39 | (5.22–5.57) | <0.001 |
| | 60–69 years | 6.03 | (4.35–8.37) | <0.001 | 6.58 | (6.32–6.85) | <0.001 |
| Education-years | No formal education | Ref | (-) | | Ref | (-) | |
| | 1–10 years | 0.63 | (0.53–0.75) | <0.001 | 0.64 | (0.63–0.65) | <0.001 |
| | 11–12 years | 0.23 | (0.15–0.35) | <0.001 | 0.28 | (0.27–0.30) | <0.001 |
| | More than 12 years | 0.67 | (0.43–1.05) | 0.080 | 0.75 | (0.72–0.79) | <0.001 |
| Working Status | Employee | Ref | (-) | | Ref | (-) | |
| | Self-employed | 1.33 | (1.08–1.64) | 0.008 | 1.28 | (1.25–1.31) | <0.001 |
| | Non-working | 0.95 | (0.75–1.20) | 0.650 | 0.92 | (0.89–0.94) | <0.001 |
| Residential area | Rural | Ref | (-) | | Ref | (-) | |
| | Urban | 0.93 | (0.79–1.09) | 0.365 | 1.03 | (1.01–1.04) | 0.012 |
| Income | Nu.0-9,000 | Ref | (-) | 0.077 | Ref | (-) | |
| | Nu.9,001–30,000 | 1.03 | (0.84–1.27) | 0.771 | 0.88 | (0.86–0.91) | <0.001 |
| | Nu.30,001–60,000 | 0.80 | (0.64–1.01) | 0.055 | 0.62 | (0.60–0.64) | <0.001 |
| | Nu.60,001- | 0.86 | (0.70–1.06) | 0.165 | 0.88 | (0.86–0.90) | <0.001 |
| Survey language | Dzongkha | Ref | (-) | | Ref | (-) | |
| | Tshanglakha | 1.69 | (1.41–2.03) | <0.001 | 1.34 | (1.32–1.37) | <0.001 |
| | Lhotshamkha | 1.24 | (1.02–1.52) | 0.032 | 1.09 | (1.07–1.12) | <0.001 |
| | English | 0.43 | (0.21–0.89) | 0.022 | 0.47 | (0.44–0.51) | <0.001 |
| Tobacco use | Never use | Ref | (-) | | Ref | (-) | |
| | Currently use | 0.79 | (0.65–0.96) | 0.016 | 0.84 | (0.82–0.85) | <0.001 |
| Alcohol consumption | Never drink | Ref | (-) | | Ref | (-) | |
| | Light or moderate drinking | 1.29 | (1.06–1.58) | 0.012 | 1.21 | (1.18–1.23) | <0.001 |
| | Heavy drinking | 1.78 | (1.43–2.22) | <0.001 | 1.43 | (1.40–1.47) | <0.001 |
| Fruit and vegetable consumption | More than 5 serves per day | Ref | (-) | | Ref | (-) | |
| | Five or fewer serves per day | 0.81 | (0.69–0.96) | 0.012 | 0.84 | (0.82–0.85) | <0.001 |
| Physical Activity | 150 min or more per week | Ref | (-) | | Ref | (-) | |
| | Less than 150 min per week | 1.00 | (0.75–1.33) | 0.987 | 1.01 | (0.98–1.05) | 0.463 |
| Salt intake | Less than 5 g per day | Ref | (-) | | Ref | (-) | |
| | 5 g or more per day | 1.05 | (0.49–2.25) | 0.905 | 1.35 | (1.22–1.50) | <0.001 |
| Blood glucose | Normal | Ref | (-) | | Ref | (-) | |
| | Abnormal | 2.92 | (2.15–3.98) | <0.001 | 2.52 | (2.43–2.61) | <0.001 |
| Total cholesterol | Lower than 240 mg/dl | Ref | (-) | | Ref | (-) | |
| | 240 mg/dl or higher | 2.66 | (1.47–4.83) | 0.001 | 3.14 | (2.93–3.37) | <0.001 |
| BMI | <18.5 | Ref | (-) | | Ref | (-) | |
| | 18.5 ≤ BMI < 25.0 | 1.36 | (0.86–2.17) | 0.194 | 1.71 | (1.62–1.80) | <0.001 |
| | 25.0 ≤ BMI < 30.0 | 2.73 | (1.70–4.39) | <0.001 | 3.27 | (3.10–3.46) | <0.001 |
| | ≥30.0 | 5.21 | (3.03–8.94) | <0.001 | 7.34 | (6.88–7.84) | <0.001 |

(*Continued*)

**Table 2.** (Continued)

| n = 1909 | | Non-weighted | | | Weighted | | |
|---|---|---|---|---|---|---|---|
| | | OR | 95%CI | p-value | AOR[1] | 95%CI | p-value |
| Family history of hypertension | Negative | Ref | (-) | | Ref | (-) | |
| | Positive | 1.14 | (0.97–1.34) | 0.110 | 1.09 | (1.07–1.11) | <0.001 |
| Cardiovascular disease | Negative | Ref | (-) | | Ref | (-) | |
| | Positive | 1.47 | (0.61–3.55) | 0.397 | 2.24 | (2.04–2.47) | <0.001 |
| Considering Suicide | Negative | Ref | (-) | | Ref | (-) | |
| | Positive | 0.69 | (0.40–1.17) | 0.163 | 0.73 | (0.69–0.78) | <0.001 |

[1] Weight-Adjusted Odds Ratio.

For sensitivity analysis (S3 Table), hypertension was redefined as a mean SBP $\geq$ 180 mmHg or DBP $\geq$ 120 mmHg and SBP $\geq$ 160 mmHg or DBP $\geq$ 100 mmHg. These trends were similar; however, the odds increased for separated, divorced, or widowed men.

## Discussion

The incidence of hypertension was significantly related to sociocultural factors in Bhutan, including survey language and gender with marital status. Similar to other countries, hypertension was also significantly related to biomedical and lifestyle factors, except tobacco use.

### Survey language

The odds of hypertension varied with the language in which the survey was administered. The highest odds of hypertension were observed for the Tshanglakha group, followed by the Lhotshamkha and Dzongkha groups, even after adjusting for other risk factors. The differences between the languages may be due to underlying cultural differences, such as dietary habits. Culture is deeply related to a person's daily life, cognition, and habits [38], incorporating differences in diet and social norms [15, 39]. The survey languages used in the study were not necessarily, but quite likely, to be identical to the respondent's native language. Language categories also correspond to regional or racial differences [40]. Such differences may influence access to vegetables and water, as well as disparities in altitude and climate [41]. Moreover, differences may exist between the language groups regarding access to hospitals and health information, or genetic difference due to race or ethnicity [18].

These findings indicate that it is necessary to strengthen the strategies for early detection and screening of hypertension, particularly in Tshanglakha and Lhotshamkha language groups, because there is a potentially hidden risk of hypertension. The survey's cross-sectional design did not enable identification of the reasons underlying this disparity, which warrants further investigation. Even within small areas such as Bhutan, careful attention should be given to cultural factors, such as regional characteristics, customs, and the environment, to enhance hypertension prevention strategies.

### Gender with marital status

The prevalence of hypertension varied by gender with marital status. Married or cohabitating women and separated, divorced, or widowed women had a higher prevalence than men, even after adjusting for biological risk factors. Other factors, such as chronic stress or unmeasured cultural factors, may also influence this observation [8, 42]. In Bhutan, being happy is

**Table 3. Multivariable logistic regression analysis for hypertension with sociocultural, lifestyle, biomedical, and mental health variables (n = 1,909).**

| n = 1909 | | Non-weighted | | | Weighted | | |
|---|---|---|---|---|---|---|---|
| | | AOR[1] | 95%CI | *p*-value | AOR[2] | 95%CI | *p*-value |
| Gender-Marital Status | Men-Married or cohabitant | Ref | (-) | | Ref | (-) | |
| | Men-Never married | 2.22 | (1.23–4.01) | 0.008 | 1.97 | (1.87–2.08) | <0.001 |
| | Men-Separated or Divorced or Widowed | 0.80 | (0.40–1.62) | 0.543 | 0.66 | (0.60–0.71) | <0.001 |
| | Women-Married or cohabitant | 1.19 | (0.92–1.54) | 0.185 | 1.27 | (1.23–1.31) | <0.001 |
| | Women-Never married | 0.57 | (0.26–1.22) | 0.146 | 0.44 | (0.40–0.49) | <0.001 |
| | Women-Separated or Divorced or Widowed | 1.35 | (0.92–1.97) | 0.123 | 1.18 | (1.12–1.25) | <0.001 |
| Age | 18–29 years | Ref | (-) | | Ref | (-) | |
| | 30–39 years | 2.14 | (1.56–2.94) | <0.001 | 2.52 | (2.43–2.60) | <0.001 |
| | 40–49 years | 3.60 | (2.57–5.04) | <0.001 | 3.48 | (3.35–3.62) | <0.001 |
| | 50–59 years | 4.78 | (3.27–7.00) | <0.001 | 5.57 | (5.33–5.82) | <0.001 |
| | 60–69 years | 7.50 | (4.75–11.85) | <0.001 | 8.56 | (8.10–9.05) | <0.001 |
| Education-years | No formal education | Ref | (-) | | Ref | (-) | |
| | 1–10 years | 0.92 | (0.72–1.18) | 0.516 | 0.96 | (0.93–0.99) | 0.003 |
| | 11–12 years | 0.52 | (0.27–1.00) | 0.048 | 0.60 | (0.57–0.64) | <0.001 |
| | More than 12 years | 0.95 | (0.46–1.95) | 0.886 | 0.95 | (0.88–1.03) | 0.212 |
| Working Status | Employee | Ref | (-) | 0.606 | Ref | (-) | |
| | Self-employed | 0.93 | (0.67–1.30) | 0.673 | 1.03 | (0.99–1.06) | 0.159 |
| | Non-working | 0.84 | (0.59–1.21) | 0.355 | 0.89 | (0.86–0.93) | <0.001 |
| Residential area | Rural | Ref | (-) | | Ref | (-) | |
| | Urban | 0.97 | (0.74–1.27) | 0.805 | 1.11 | (1.07–1.14) | <0.001 |
| Income | Nu.0-9,000 | Ref | (-) | | Ref | (-) | |
| | Nu.9,001–30,000 | 1.11 | (0.85–1.46) | 0.457 | 0.90 | (0.88–0.93) | <0.001 |
| | Nu.30,001–60,000 | 1.07 | (0.78–1.48) | 0.671 | 0.73 | (0.70–0.76) | <0.001 |
| | Nu.60,001- | 1.18 | (0.85–1.65) | 0.327 | 1.05 | (1.01–1.09) | 0.015 |
| Survey language | Dzongkha | Ref | (-) | | Ref | (-) | |
| | Tshanglakha | 1.72 | (1.32–2.25) | <0.001 | 1.24 | (1.20–1.27) | <0.001 |
| | Lhotshamkha | 1.37 | (1.05–1.79) | 0.019 | 1.09 | (1.06–1.12) | <0.001 |
| | English | 0.31 | (0.10–1.00) | 0.05 | 0.29 | (0.26–0.32) | <0.001 |
| Tobacco use | Never use | Ref | (-) | | Ref | (-) | |
| | Currently use | 0.75 | (0.57–0.98) | 0.037 | 0.82 | (0.79–0.84) | <0.001 |
| Alcohol consumption | Never drink | Ref | (-) | | Ref | (-) | |
| | Light or moderate drinking | 1.27 | (1.00–1.62) | 0.054 | 1.24 | (1.20–1.27) | <0.001 |
| | Heavy drinking | 1.55 | (1.18–2.04) | 0.002 | 1.32 | (1.28–1.36) | <0.001 |
| Fruit and vegetable consumption | More than 5 serves per day | Ref | (-) | | Ref | (-) | |
| | Five or fewer serves per day | 1.09 | (0.87–1.37) | 0.472 | 0.96 | (0.94–0.99) | 0.003 |
| Physical Activity | 150 min or more per week | Ref | (-) | | Ref | (-) | |
| | Less than 150 min per week | 0.80 | (0.54–1.19) | 0.272 | 0.82 | (0.78–0.86) | <0.001 |
| Salt intake | Below 5 g per day | Ref | (-) | | Ref | (-) | |
| | 5 g or more per day | 1.40 | (0.55–3.58) | 0.479 | 1.41 | (1.24–1.60) | <0.001 |
| Blood glucose | Normal | Ref | (-) | | Ref | (-) | |
| | Abnormal | 2.13 | (1.43–3.17) | <0.001 | 2.27 | (2.16–2.37) | <0.001 |
| Total cholesterol | Lower than 240 mg/dl | Ref | (-) | | Ref | (-) | |
| | 240 mg/dl or higher | 1.31 | (0.64–2.70) | 0.456 | 1.66 | (1.52–1.81) | <0.001 |

(*Continued*)

**Table 3.** (Continued)

| n = 1909 | | Non-weighted | | | Weighted | | |
|---|---|---|---|---|---|---|---|
| | | AOR[1] | 95%CI | *p*-value | AOR[2] | 95%CI | *p*-value |
| BMI | <18.5 | Ref | (-) | | Ref | (-) | |
| | 18.5 ≤ BMI < 25.0 | 1.53 | (0.85–2.76) | 0.156 | 1.79 | (1.68–1.92) | <0.001 |
| | 25.0 ≤ BMI < 30.0 | 2.61 | (1.42–4.78) | 0.002 | 2.95 | (2.75–3.16) | <0.001 |
| | ≥30.0 | 3.64 | (1.84–7.21) | <0.001 | 4.23 | (3.91–4.57) | <0.001 |
| Family history of hypertension | Negative | Ref | (-) | | Ref | (-) | |
| | Positive | 1.35 | (1.09–1.69) | 0.007 | 1.13 | (1.11–1.16) | <0.001 |
| Cardiovascular disease | Negative | Ref | (-) | | Ref | (-) | |
| | Positive | 0.88 | (0.29–2.66) | 0.826 | 0.65 | (0.56–0.76) | <0.001 |
| Considering Suicide | Negative | Ref | (-) | | Ref | (-) | |
| | Positive | 0.79 | (0.38–1.66) | 0.528 | 1.24 | (1.14–1.35) | <0.001 |

[1] Adjusted with all explanatory variables.

[2] Weight adjusted and adjusted with all explanatory variables.

considered a virtue [15], which from a social background perspective might make it more difficult to verbalize negative emotions. However, verbalizing negative emotions can help prevent people from being stressed [43]. Therefore, strategies that enable vulnerable people to reduce stress and negative emotions may be necessary.

Previous studies in other countries indicate that divorced individuals or those who have lost a partner have higher health risks [44]. Furthermore, divorced men or those who have lost a partner have poorer mental health than women, and marital status satisfaction and social roles influence health [45, 46]. Specifically, hypertension is linked to chronic stress [47], whereas unmeasured factors related to gender with marital status, such as sociocultural vulnerability and gender issues, may also influence health [47, 48]. Further research is necessary to confirm whether women have a sociocultural vulnerability towards hypertension compared to men in Bhutan. When promoting the early detection and screening of hypertension, careful attention should be given to married or cohabitating women and separated, divorced, or widowed women in Bhutan.

## Biomedical and lifestyle factors

Biological factors, such as aging, non-standard BMI, impaired fasting blood glucose, and high levels of total blood cholesterol, were risk factors of hypertension that are common across countries [19]. In contrast, tobacco use was not identified as a risk factor in the present study. These risk factors also influence the incidence of cardiovascular diseases [3]. Periodic screening and the early detection of hypertension may help prevent cardiovascular disease. As a mental health factor, suicide ideation was also significantly associated with hypertension, similar to findings in other countries. As such, periodic blood pressures assessments are recommended when monitoring mental health [1, 2]. Although people might hesitate to ask for help concerning mental health issues (such as suicide attempts), a healthcare worker may identify concerns about a patient's mental health when monitoring their blood pressure and may serve as gatekeepers for detection and prevention. Conversely, no consistent significant association was found between hypertension and low levels of education or working status. This finding may indicate that other biological or sociocultural factors have a strong influence on hypertension.

In terms of behavior, reducing alcohol consumption and avoiding excessive salt intake have been identified as important primary prevention strategies by previous studies in Bhutan [13,

**Table 4. Multivariable logistic regression analysis for hypertension with sociocultural, lifestyle, biomedical, and mental health variables for participants aged 40 years or more.**

| n = 921 | | Non-weighted | | | Weighted | | |
|---|---|---|---|---|---|---|---|
| | | AOR[1] | 95%CI | p-value | AOR[2] | 95%CI | p-value |
| Gender-Marital Status | Men-Married or cohabitant | Ref | (-) | | Ref | (-) | |
| | Men-Never married | 1.06 | (0.18–6.10) | 0.952 | 0.48 | (0.39–0.59) | <0.001 |
| | Men-Separated or Divorced or Widowed | 0.90 | (0.41–1.97) | 0.785 | 0.62 | (0.57–0.69) | <0.001 |
| | Women-Married or cohabitant | 1.58 | (1.11–2.26) | 0.012 | 1.88 | (1.79–1.97) | <0.001 |
| | Women-Never married | 1.29 | (0.32–5.13) | 0.721 | 0.81 | (0.62–1.04) | 0.098 |
| | Women-Separated or Divorced or Widowed | 1.71 | (1.07–2.73) | 0.024 | 1.81 | (1.68–1.95) | <0.001 |
| Age | 40–49 years | Ref | (-) | <0.001 | Ref | (-) | |
| | 50–59 years | 1.45 | (1.04–2.01) | 0.028 | 1.76 | (1.68–1.84) | <0.001 |
| | 60–69 years | 2.39 | (1.56–3.66) | <0.001 | 3.08 | (2.90–3.27) | <0.001 |
| Education-years | No formal education | Ref | (-) | 0.276 | Ref | (-) | |
| | 1–10 years | 1.07 | (0.72–1.60) | 0.729 | 1.29 | (1.22–1.36) | <0.001 |
| | 11–12 years | 3.94 | (0.36–42.63) | 0.259 | 4.53 | (3.43–5.97) | <0.001 |
| | More than 12 years | 3.75 | (0.80–17.70) | 0.095 | 7.01 | (5.53–8.89) | <0.001 |
| Working Status | Employee | Ref | (-) | 0.793 | Ref | (-) | |
| | Self-employed | 0.87 | (0.52–1.47) | 0.609 | 1.01 | (0.94–1.07) | 0.883 |
| | Non-working | 0.82 | (0.46–1.45) | 0.496 | 0.90 | (0.83–0.97) | 0.005 |
| Residential area | Rural | Ref | (-) | | Ref | (-) | |
| | Urban | 0.86 | (0.56–1.30) | 0.472 | 1.01 | (0.96–1.07) | 0.659 |
| Income | Nu.0-9,000 | Ref | (-) | | Ref | (-) | |
| | Nu.9,001–30,000 | 1.19 | (0.83–1.71) | 0.349 | 1.13 | (1.07–1.18) | <0.001 |
| | Nu.30,001–60,000 | 1.51 | (0.96–2.37) | 0.073 | 1.16 | (1.09–1.24) | <0.001 |
| | Nu.60,001- | 1.19 | (0.74–1.93) | 0.47 | 1.33 | (1.24–1.42) | <0.001 |
| Survey language | Dzongkha | Ref | (-) | | Ref | (-) | |
| | Tshanglakha | 1.80 | (1.22–2.66) | 0.003 | 1.41 | (1.33–1.49) | <0.001 |
| | Lhotshamkha | 1.45 | (0.99–2.12) | 0.056 | 1.13 | (1.07–1.19) | <0.001 |
| | English | 0.51 | (0.10–2.80) | 0.442 | 0.27 | (0.22–0.35) | <0.001 |
| Tobacco use | Never use | Ref | (-) | | Ref | (-) | |
| | Currently use | 0.68 | (0.46–1.00) | 0.049 | 0.62 | (0.59–0.65) | <0.001 |
| Alcohol consumption | Never drink | Ref | (-) | | Ref | (-) | |
| | Light or moderate drinking | 1.40 | (0.98–1.98) | 0.063 | 1.43 | (1.36–1.49) | <0.001 |
| | Heavy drinking | 1.91 | (1.29–2.83) | 0.001 | 2.25 | (2.13–2.37) | <0.001 |
| Fruit and vegetable consumption | More than 5 serves per day | Ref | (-) | | Ref | (-) | |
| | 5 or fewer serves per day | 1.15 | (0.82–1.60) | 0.418 | 1.21 | (1.16–1.27) | <0.001 |
| Physical Activity | 150 min or more per week | Ref | (-) | | Ref | (-) | |
| | Less than 150 min per week | 0.91 | (0.50–1.68) | 0.772 | 0.89 | (0.82–0.97) | 0.008 |
| Salt intake | Less than 5 g per day | Ref | (-) | | Ref | (-) | |
| | 5 g per day or more | 6.02 | (1.20–30.18) | 0.029 | 14.93 | (10.95–20.36) | <0.001 |
| Blood glucose | Normal | Ref | (-) | | Ref | (-) | |
| | Abnormal | 2.28 | (1.37–3.81) | 0.002 | 2.88 | (2.67–3.11) | <0.001 |
| Total cholesterol | Lower than 240 mg/dl | Ref | (-) | | Ref | (-) | |
| | 240 mg/dl or higher | 0.74 | (0.30–1.85) | 0.524 | 0.95 | (0.84–1.08) | 0.455 |
| BMI | <18.5 | Ref | (-) | | Ref | (-) | |
| | 18.5 ≤ BMI < 25.0 | 1.26 | (0.59–2.70) | 0.548 | 0.88 | (0.79–0.97) | 0.009 |
| | 25.0 ≤ BMI < 30.0 | 2.19 | (0.99–4.86) | 0.054 | 1.44 | (1.29–1.60) | <0.001 |
| | ≥30.0 | 3.48 | (1.38–8.81) | 0.008 | 2.05 | (1.82–2.32) | <0.001 |

*(Continued)*

**Table 4.** (Continued)

| n = 921 | | Non-weighted | | | Weighted | | |
|---|---|---|---|---|---|---|---|
| | | AOR[1] | 95%CI | p-value | AOR[2] | 95%CI | p-value |
| Family history of hypertension | Negative | Ref | (-) | | Ref | (-) | |
| | Positive | 1.36 | (0.98–1.89) | 0.064 | 1.38 | (1.32–1.44) | <0.001 |
| Cardiovascular disease | Negative | Ref | (-) | | Ref | (-) | |
| | Positive | 3.13 | (0.60–16.42) | 0.177 | 1.12 | (0.92–1.36) | 0.255 |
| Considering Suicide | Negative | Ref | (-) | | Ref | (-) | |
| | Positive | 1.16 | (0.35–3.87) | 0.81 | 3.32 | (2.74–4.03) | <0.001 |

[1] Adjusted with all explanatory variables.

[2] Weight adjusted and adjusted with all explanatory variables.

15]. Similar to other countries [19], alcohol consumption and excess salt intake are risk factors for hypertension in Bhutan [1, 2]. In urban areas, people have moved from various regions, resulting in communities of mixed languages. However, in rural areas, each village has its own native language. In Bhutan, chili peppers, oil, and salt are usually served as side dishes, a practice which may be due to environmental factors, such as the high altitude, which prevents the growth of fruits and vegetables, and the need to preserve food for a long time. People may find it difficult to adjust to rapid social change in modern society. It is also possible that people may miss the foods they ate in their childhood, even after moving to urban areas and changing their lifestyle. The supplementary materials (S5 Table) report salt intake by survey language.

In the present study, the odds of hypertension significantly decreased with any current use of tobacco. Although tobacco is widely known to be harmful to health [19], tobacco use in Bhutan may protect from hypertension as people tend to smoke in a social context, i.e., when chatting together, which symbolizes friendship in terms of social exchange [15]. As Buddhists comprise the majority of Bhutan's population, religious or public places remain tobacco-free [49]. In 2004, the Government of Bhutan banned the sale of tobacco products in the country or made them available with 100% tax [50]. Tobacco consumption may be suppressed due to these inhibitory measures, and the health-promoting effects of social interaction may outweigh the harm from tobacco use.

## Strengths and limitations

This study had several strengths and limitations. The study design was cross-sectional, which renders it difficult to establish a cause-and-effect relationship. The dataset from the "National survey for noncommunicable disease risk factors and mental health using WHO STEPS approach in Bhutan-2014" is representative of the entire population of Bhutan with a valid response rate of 97%. However, there is the possibility of non-response bias. Accordingly, we also conducted multiple imputation analysis (S4 Table), which found the same trends. In addition, some self-reported variables are exposed to the possibility of bias. For example, socially desirable answers may have influenced the results of face-to-face interviews. However, the bias was minimized, because trained health professional staff did the survey under supervision. Interviews were the only suitable method for collecting data from this target population with low literacy rates.

In terms of feasibility, the survey utilized the best means possible to conduct such an extensive survey in Bhutan, which included specific biomarkers. An advantage of the study method was that it collected many objective variables, such as blood pressure, blood glucose, total cholesterol, and spot urine. Ideally, 24-hour urine analysis would have been more accurate for salt

intake than spot urine, and triglyceride, high-density lipoprotein, and low-density lipoprotein levels would have been more accurate than total cholesterol as a risk factor for hypertension. Nonetheless, the study is one of few in Bhutan that has focused on social and cultural factors related to hypertension after adjusting for all possible and available confounders.

## Conclusion

This study indicates that the prevalence of hypertension in Bhutan was associated with cultural factors such as regional characteristics, customs, environment, and gender with and marital status. The screening and early detection of hypertension should be strongly emphasized, particularly among people who speak Tshanglakha, Lhotshamkha, and women who are married, separated, divorced, or widowed. The mechanism by which cultural differences influence hypertension in Bhutan should be further investigated.

## Supporting information

**S1 File. Sample size and survey procedures quoted from the original report of the World Health Organization and Ministry of Health in Bhutan (National survey for noncommunicable disease risk factors and mental health using WHO STEPS approach in Bhutan—2014).**
(PDF)

**S1 Fig. Interaction effect between gender and marital status (Hypertension\*Gender).**
(TIF)

**S2 Fig. Interaction effect between gender and marital status (Hypertension\*Marital Status).**
(TIF)

**S3 Fig. Interaction effect between gender and marital status (Hypertension\*Gender\*Marital Status).**
(TIF)

**S1 Table. Multivariable logistic regression analysis for hypertension with sociocultural, lifestyle, biomedical, and mental health variables: Without considering interaction effects (n = 1,909).**
(DOCX)

**S2 Table. Multivariable logistic regression analysis for hypertension with sociocultural, lifestyle, biomedical, and mental health variables: Including interaction effects (n = 1,909).**
(DOCX)

**S3 Table. Multivariable logistic regression analysis for Hypertension with Sociocultural, Lifestyle, Biomedical, and mental health variables: High blood pressure was defined as being diagnosed or under medication for hypertension, or a mean SBP ≥180 mmHg or DBP≥120 mmHg, and SBP≧160 mmHg or DBP≧100 mmHg, according to the severity of hypertension categories.**
(DOCX)

**S4 Table. Multivariable logistic regression analysis for hypertension with sociocultural, lifestyle, biomedical, and mental health variables: Multiple imputation.**
(DOCX)

**S5 Table. Estimated salt intake by survey language category.** Estimated Salt Intake per day (g) by Survey language.
(DOCX)

## Acknowledgments

We appreciate all participants in Bhutan. We would like to thank Cambridge English Correction Service for the English language review. We appreciate the Khesar Gyalpo University of Medical Sciences of Bhutan and the Ministry of Health of Bhutan as well as all the support extended by the Graduate School of Medicine, Kyoto University, Japan.

## Author Contributions

**Conceptualization:** Hiromi Kohori Segawa, Hironori Uematsu, Nidup Dorji, Ugyen Wangdi, Pemba Yangchen, Ryota Sakamoto, Yuichi Imanaka.

**Data curation:** Hiromi Kohori Segawa, Pemba Yangchen.

**Formal analysis:** Hiromi Kohori Segawa.

**Funding acquisition:** Hiromi Kohori Segawa, Yuichi Imanaka.

**Investigation:** Hiromi Kohori Segawa.

**Methodology:** Hiromi Kohori Segawa, Hironori Uematsu, Susumu Kunisawa.

**Project administration:** Hiromi Kohori Segawa, Nidup Dorji, Ugyen Wangdi, Chencho Dorjee.

**Supervision:** Chencho Dorjee, Yuichi Imanaka.

**Visualization:** Hiromi Kohori Segawa.

**Writing – original draft:** Hiromi Kohori Segawa.

**Writing – review & editing:** Hiromi Kohori Segawa, Hironori Uematsu, Nidup Dorji, Susumu Kunisawa, Ryota Sakamoto, Yuichi Imanaka.

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
