## [Decision Letter · Decision Letter 0]

4 Jun 2021

PONE-D-21-08522

Marital status, cultural differences, and vulnerability to hypertension : Findings from the national survey for noncommunicable disease risk factors and mental health using WHO STEPS in Bhutan

PLOS ONE

Dear Dr. Imanaka,

Thank you for submitting your manuscript to PLOS ONE. After careful consideration, we feel that it has merit but does not fully meet PLOS ONE’s publication criteria as it currently stands. Therefore, we invite you to submit a revised version of the manuscript that addresses the points raised during the review process.

We look forward to receiving your revised manuscript.

Kind regards,

Akihiro Nishi, M.D., Dr.P.H.

Academic Editor

PLOS ONE

Journal Requirements:

2.Please provide additional details regarding participant consent. In the ethics statement in the Methods and online submission information, please ensure that you have specified (1) whether consent was informed and (2) what type you obtained (for instance, written or verbal, and if verbal, how it was documented and witnessed). If your study included minors, state whether you obtained consent from parents or guardians. If the need for consent was waived by the ethics committee, please include this information.

3.We note that you have indicated that data from this study are available upon request. PLOS only allows data to be available upon request if there are legal or ethical restrictions on sharing data publicly. For information on unacceptable data access restrictions, please see http://journals.plos.org/plosone/s/data-availability#loc-unacceptable-data-access-restrictions.

Additional Editor Comments:

Please follow most of the Reviewer 1's suggestions. The editor hopes that the manuscript is improved at the second round.

Reviewers' comments:

Reviewer's Responses to Questions

**Comments to the Author**

1. Is the manuscript technically sound, and do the data support the conclusions?

Reviewer #1: Yes

Reviewer #2: Partly

2. Has the statistical analysis been performed appropriately and rigorously? 

Reviewer #1: Yes

Reviewer #2: No

3. Have the authors made all data underlying the findings in their manuscript fully available?

Reviewer #1: No

Reviewer #2: No

4. Is the manuscript presented in an intelligible fashion and written in standard English?

Reviewer #1: Yes

Reviewer #2: No

5. Review Comments to the Author

Reviewer #1: The current manuscript aimed to examine the association of gender with marital status, language, and hypertension using multivariate logistic regression analysis. The findings demonstrated that women had higher odds ratio than men, during marriage, and separated, divorced, or widowed women. Language wise, Tshanglakha or Lhotshamkha scored high odds. I believe that the topic is interesting for the scientific community, however the manuscript still needs some work for publication.

1. Report on state of the art and comparable research is a question to evaluate innovations of presented research.

Tashi Dendup et al (2020) Risk factors associated with hypertension in Bhutan: findings from the National Health Survey. Journal of Health Research.

2. Please add the definition of hypertension used in the subgroup analysis for age 40 and over to the dependent variable in the definition of variable. (p9, lines:168-170)

3. Make sure the correlation coefficient between marital status and age is not too high.

4. The note is missing. -Table1. 1) and Table 4. 4)

Reviewer #2: This study aimed to assess the risk factors for hypertension in Bhutan,

I have some comments:

1- Extensive English language editing is needed.

2- Since this study aimed to detect the risk factors for HTN, physicians in Bhutan should use the results of this study to detect individuals at high risk for HTN; this is a primary prevention.

3- More details about the data collecting tool and methods to calculate salt intake are needed.

4- More details about HTN measurement should be given (possible calibration, how many readings, self-reported or made by HCWs, rest before measurement, etc.)

5- Sampling process and steps should be provided. It is not enough to mention the name of the sampling method.

6- Formal sample size calculation is needed.

7- Response rate and causes of nonresponse should be provided.

8- Since a random sampling approach was used, I do not think that weighted analysis is needed.

9- I do not think that combining gender and marital status in one variable is justified. They should be separated.

10- The language of the survey cannot be a risk factor, per se, but indication of the culture or background of the participants who used this language. This should be clarified.

11- The discussion lacks the national perspective. In other words, all these risk factors are well-known; the discussion has to be deeper and focus on sociodemographic and cultural habits that specify Bhutan not simple comparisons with other studies.

12- I do not think that categorizing fruits and vegetables to 5 or more dishes per day is optimal. Maybe per week; please check or make more categories. You cannot advise people to eat more than 5 dishes of fruits and vegetables per day.

13- Also, how big was the dish?

14- I highly recommend using the regression model to make a national hypertension score; it will have epidemiological and clinical implications.

15- limitation sections should include the cross-sectional design, not including many risk factors such as dyslipidemia, all variables were self-reported, and possibility of nonresponse bias. These complications should be discussed in detail.

16- Most biostatisticians advise against complete case analysis. It minimizes the study power and can significantly change the results, especially when many factors are included in the model which is the situation in this study. Multiple imputation should be done.

6. PLOS authors have the option to publish the peer review history of their article (what does this mean?). If published, this will include your full peer review and any attached files.

Reviewer #1: No

Reviewer #2: **Yes: **Ahmed Arafa

---

## [Author Response · Author response to Decision Letter 0]

10 Aug 2021

Dear Editor and reviewers,

We really appreciate your review and feedback during these difficult times of the COVID-19 pandemic. We have improved our manuscript by responding to your comments, as detailed below:

⇒We have addressed PLOS ONE's style requirements, including those for file naming.

2.Please provide additional details regarding participant consent. In the ethics statement in the Methods and online submission information, please ensure that you have specified (1) whether consent was informed and (2) what type you obtained (for instance, written or verbal, and if verbal, how it was documented and witnessed). If your study included minors, state whether you obtained consent from parents or guardians. If the need for consent was waived by the ethics committee, please include this information.

⇒We have added an explanation of the ethical considerations of our study, as detailed below:

(p11, lines:189-195)　 The Research Ethics Boards of Health, Ministry of Health of the Royal Government of Bhutan (No. REBH/Approval/2018/089), and the Ethics Committee of the Kyoto University Hospital and Graduate School of Medicine (No. R1796) approved the study protocol. The above ethical committees waived the requirement for individual informed consent as this secondary analysis only used de-identified data. We have submitted a written pledge of confidentiality to the Ministry of Health of the Royal Government of Bhutan.

3.We note that you have indicated that data from this study are available upon request. PLOS only allows data to be available upon request if there are legal or ethical restrictions on sharing data publicly. For information on unacceptable data access restrictions, please see http://journals.plos.org/plosone/s/data-availability#loc-unacceptable-data-access-restrictions.

⇒We updated the data availability statement by adding contact information for the Ministry of Health, Bhutan, which is the authority for the data.

(p26, lines:403-404) Contact :P.O. Box: 726, Kawajangsa,Thimphu, Bhutan Phone No: +975-2-328095, 321842 , 322602, 328091

⇒We have added captions in the supporting information files.

Reviewer's Responses to Questions

Comments to the Author

1. Is the manuscript technically sound, and do the data support the conclusions?

Reviewer #1: Yes

Reviewer #2: Partly

2. Has the statistical analysis been performed appropriately and rigorously?

Reviewer #1: Yes

Reviewer #2: No

3. Have the authors made all data underlying the findings in their manuscript fully available?

Reviewer #1: No

Reviewer #2: No

⇒　We added the information below: 

(p26, lines:396-404) Since the Ministry of Health, Bhutan is the authority for the data, the authors were not able to make it available with this manuscript. However, the original report (World Health Organization and Ministry of Health in Bhutan. National survey for noncommunicable disease risk factors and mental health using WHO STEPS approach in Bhutan – 2014) is available from https://apps.who.int/iris/handle/10665/204659. Requests to access the raw data can be directed to the Research Ethics Board of Health at the Royal Government of Bhutan or the Ministry of Health, Bhutan.

Contact :P.O. Box: 726, Kawajangsa, Thimphu, Bhutan Phone No: +975-2-328095, 321842 , 322602, 328091

4. Is the manuscript presented in an intelligible fashion and written in standard English?

Reviewer #1: Yes

Reviewer #2: No

⇒English is our second language. So we asked a professional English copyeditor to review and edit our manuscript to ensure it is written in standard English.

5. Review Comments to the Author

Reviewer #1: The current manuscript aimed to examine the association of gender with marital status, language, and hypertension using multivariate logistic regression analysis. The findings demonstrated that women had higher odds ratio than men, during marriage, and separated, divorced, or widowed women. Language wise, Tshanglakha or Lhotshamkha scored high odds. I believe that the topic is interesting for the scientific community, however the manuscript still needs some work for publication.

1. Report on state of the art and comparable research is a question to evaluate innovations of presented research.

Tashi Dendup et al (2020) Risk factors associated with hypertension in Bhutan: findings from the National Health Survey. Journal of Health Research.

⇒　Thank you for this suggestion. Although we did not mention it, the paper you identified has significance for our research theme. The paper was based on a 2012 living standards survey in Bhutan, which reported only self-reported hypertension. However, our paper was based on diagnosed cases of hypertension from the STEP survey in Bhutan (2014). Also, we focused on the differences in risk according to culture or social norms. We added the following sentence to the introduction:

→ (p5, lines:89-92) However, to the best of our knowledge, there have been few studies in Bhutan investigating issues related to preventing hypertension, such as examining social and cultural factors. Specifically, investigations using objective variables (such as blood pressure measurement) in surveys or biomarkers are scarce in Bhutan.

2. Please add the definition of hypertension used in the subgroup analysis for age 40 and over to the dependent variable in the definition of variable. (p9, lines:168-170)

⇒　Thank you for this suggestion. We have now included the definition of hypertension for the subgroup analyses. 

→　(p10-11, lines:182-188) Hypertension in subgroup analyses was defined by the following criteria: (1) systolic blood pressure (SBP) ≥ 140 mmHg or diastolic blood pressure (DBP) ≥ 90 mmHg, measured as the average of three measurements taken by health staff in the survey; (2) a previous diagnosis of hypertension by healthcare workers; (3) currently taking medication for hypertension [19].

3. Make sure the correlation coefficient between marital status and age is not too high.

⇒　Thank you for this comment. A significant weak correlation (r=0.34) was found between age and marital status (ordinal variable: Never married＝1, Currently married or cohabiting＝2, Separated, divorced, or widowed＝3). It is not too high.

→（p,17, line:258）The correlation coefficient between marital status and age was not too high (r=0.34).

4. The note is missing. -Table1. 1) and Table 4. 4)

⇒　Thanks so much for this observation. We have inserted the missing footnote into Table 1, and we have erased point 4 from Table 4.

Reviewer #2: This study aimed to assess the risk factors for hypertension in Bhutan,

I have some comments:

1- Extensive English language editing is needed.

⇒　English is our second language. We asked a professional English copyeditor to review and edit our manuscript to ensure it is written in standard English.

2- Since this study aimed to detect the risk factors for HTN, physicians in Bhutan should use the results of this study to detect individuals at high risk for HTN; this is a primary prevention.

⇒　Thank you for this comment. As pointed out, globally, early detection and prevention of hypertension may be classified as primary prevention. Aspects of the primary prevention of hypertension, such as a healthy diet and proper exercise, have been studied and are being addressed in Bhutan. To build on these strategies, we have focused on the association between sociocultural background and the early detection of hypertension, so we designated this aspect as secondary in the paper. However, as you pointed out, this may confuse some readers, so we have changed the term “secondary prevention” to “further prevention strategies.” 

3- More details about the data collecting tool and methods to calculate salt intake are needed.

⇒　Thanks for your comment. This study is a secondary analysis of previously collected data. The detailed survey procedure and data collection tool are available in the original report (World Health Organization and Ministry of Health in Bhutan. National survey for noncommunicable disease risk factors and mental health using WHO STEPS approach in Bhutan – 2014).

We have confirmed the details below:

P5, “World Health Organization and Ministry of Health in Bhutan. National survey for noncommunicable disease risk factors and mental health using WHO STEPS approach in Bhutan – 2014.”

“The generic WHO STEPS survey protocol and tools were adapted to local conditions in Bhutan.　A container was provided to collect urine samples prior to the beginning of the fast. Participants were asked to go to the testing centre set up by the survey team (located in the vicinity) the next morning. Here the blood samples were taken and the urine samples delivered to the survey team. Urine samples were sent by the survey team to the Jigme Dorji Wangchuck National Referral Hospital Laboratory (JDWNRH) in Thimphu for analysis of sodium and creatinine to determine mean population salt intake.”

Details of the method to estimate salt intake are now included in the manuscript. Salt intake was calculated using the Tanaka Formula (Tanaka T, Okamura T, Miura K, Kadowaki T, Ueshima H, Nakagawa H, et al. A simple method to estimate populational 24-h urinary sodium and potassium excretion using a casual urine specimen. Journal of Human Hypertension. 2002;16(2):97-103).

Salt Intake per day(g) = (21.98 * ((Spot Urinary Sodium/

(Spot Urinary Creatinine *10/0.0884)) * (((14.89* Weight (Kg))+(16.14*height(m))*(2.04*age))-2244.45))**0.392)/17.1

4- More details about HTN measurement should be given (possible calibration, how many readings, self-reported or made by HCWs, rest before measurement, etc.)

⇒　Thanks for this comment. We added an explanation of how the variables were defined (p7, lines:127-130). We used the average of three measurements taken in the survey.

P31 “World Health Organization and Ministry of Health in Bhutan. National survey for noncommunicable disease risk factors and mental health using WHO STEPS approach in Bhutan – 2014.”

blood pressure of the respondents (who consented) was also measured by trained health-care workers. 

5- Sampling process and steps should be provided. It is not enough to mention the name of the sampling method.

⇒　Thanks for your comment. We added a supporting information section (S1) and have provided more detail in the data source section of the manuscript. We also detail the process in figure 1.

6- Formal sample size calculation is needed.

⇒　Thank you for your valuable suggestion. We added the sample size calculations to the supporting information section (S1)

7- Response rate and causes of nonresponse should be provided.

⇒　Thanks for your comment. As you suggested, we added this information to the supporting information section (S1) 

8- Since a random sampling approach was used, I do not think that weighted analysis is needed.

⇒　Thank you for this suggestion. We understand your point, and so we have now reported the results of the crude analysis (non-weighted adjustment). We would like to re-confirm that the study did not use simple random sampling. Rather, it was a multistage random sampling technique, the outcomes of which were considered representativeness of the nation. Hence, we now report both the crude (non-weighted adjustment) and the weighted analysis. 

9- I do not think that combining gender and marital status in one variable is justified. They should be separated.

⇒　Thank you for your comment. We completely agree with your comment and accept the short explanation for this. There was an interaction effect between gender and marital status. When we used each variable independently in the model, there was no significant effect. However, there was a significant difference when we added the interaction term. In order to show the interaction effect, we combined gender and marital status in this study. For reference, we show the analysis using sole independent variables and the analysis with the interaction term in the supporting information (S2).

10- The language of the survey cannot be a risk factor, per se, but indication of the culture or background of the participants who used this language. This should be clarified.

⇒　Thanks for your comment. We improved the explanation in the discussion.

(p20, lines:292-297) The differences between the languages may be due to underlying cultural differences, such as dietary habits. Culture is deeply related to a person’s daily life, cognition, and habits [38], incorporating differences in diet and social norms [15,39]. The survey languages used in the study were not necessarily, but quite likely, to be identical to the respondent’s native language. Language categories also correspond to regional or racial differences [40]. 

11- The discussion lacks the national perspective. In other words, all these risk factors are well-known; the discussion has to be deeper and focus on sociodemographic and cultural habits that specify Bhutan not simple comparisons with other studies.

⇒　Thanks a lot for your meaningful comments. We added 2 points to the discussion section.

１）(p21, lines:310-314) In Bhutan, being happy is considered a virtue [15], which from a social background perspective might make it more difficult to verbalize negative emotions. However, verbalizing negative emotions can help prevent people from being stressed [43].

2) (p22-23, lines:341-348) In urban areas, people have moved from various regions, resulting in communities of mixed languages. However, in rural areas, each village has its own native language. In Bhutan, chili peppers, oil, and salt are usually served as side dishes, a practice which may be due to environmental factors, such as the high altitude, which prevents the growth of fruits and vegetables, and the need to preserve food for a long time. People may find it difficult to adjust to rapid social change in modern society. It is also possible that people may miss the foods they ate in their childhood, even after moving to urban areas and changing their lifestyle. The supplementary materials (S 5) report salt intake by survey language.

12- I do not think that categorizing fruits and vegetables to 5 or more dishes per day is optimal. Maybe per week; please check or make more categories. You cannot advise people to eat more than 5 dishes of fruits and vegetables per day.

13- Also, how big was the dish?

⇒　Thanks for your comments. We would like to argue here, that the WHO recommends more than 5 serves of fruits and vegetables per day. The WHO provided the survey manual according to previous literature reviews. In Japan, we recommend greater vegetable consumption than WHO guidelines.

As you mentioned, the serving size is also very important. The report stated that “the quantity of intake was measured by servings: one serving of fruit was defined as equal to a medium-sized banana or apple or equivalent, and one serving of vegetables was equal to one cup of green leafy vegetables or half a cup of cooked vegetables.” We added these definitions to the manuscript. 

 I (Hiromi Kohori-Segawa) conducted a field survey in 2017, and at that time, I daily photographed servings of food on a piece of grid paper in rural and urban Bhutan. I found two categories of participants; those who were conscious of their intake of fruits and vegetables and those who were not. I feel it is unnecessary to categorize servings into more detail at the present time. 

14- I highly recommend using the regression model to make a national hypertension score; it will have epidemiological and clinical implications.

⇒　Thank you for your valuable advice. We will try to include health resource data and make identifying a national hypertension score our next step in researching this important issue. 

15- limitation sections should include the cross-sectional design, not including many risk factors such as dyslipidemia, all variables were self-reported, and possibility of nonresponse bias. These complications should be discussed in detail.

⇒　Thanks for your comment. As you suggested, we added 

(p23-24, lines:358-375) This study had several strengths and limitations. The study design was cross-sectional, which renders it difficult to establish a cause-and-effect relationship. The dataset from the “National survey for noncommunicable disease risk factors and mental health using WHO STEPS approach in Bhutan-2014” is representative of the entire population of Bhutan with a valid response rate of 97%. However, there is the possibility of non-response bias. Accordingly, we also conducted multiple imputation analysis (S 4), which found the same trends. In addition, some self-reported variables are exposed to the possibility of bias. For example, socially desirable answers may have influenced the results of face-to-face interviews. However, the bias was minimized, because trained health professional staff did the survey under supervision. Interviews were the only suitable method for collecting data from this target population with low literacy rates. 

In terms of feasibility, the survey utilized the best means possible to conduct such an extensive survey in Bhutan, which included specific biomarkers. An advantage of the study method was that it collected many objective variables, such as blood pressure, blood glucose, total cholesterol, and spot urine. Ideally, 24-hour urine analysis would have been more accurate for salt intake than spot urine, and triglyceride, high-density lipoprotein, and low-density lipoprotein levels would have been more accurate than total cholesterol as a risk factor for hypertension. Nonetheless, the study is one of few in Bhutan that has focused on social and cultural factors related to hypertension after adjusting for all possible and available confounders.

16- Most biostatisticians advise against complete case analysis. It minimizes the study power and can significantly change the results, especially when many factors are included in the model which is the situation in this study. Multiple imputation should be done.

⇒　Thanks so much for your comments on our statistical analysis. Although it was not shown in the manuscript, no obvious bias was shown when we conducted missing value analysis. As you pointed out, it is difficult to prove whether the missing values were random or otherwise. We also agree theoretically that it is better to do multiple imputation. We have attached the results of the multiple imputation analysis (S 4), which shows similar trends to the complete case analysis.

6. PLOS authors have the option to publish the peer review history of their article (what does this mean?). If published, this will include your full peer review and any attached files.

Do you want your identity to be public for this peer review? For information about this choice, including consent withdrawal, please see our Privacy Policy.

Reviewer #1: No

Reviewer #2: Yes: Ahmed Arafa

Authors: Yes

---

## [Decision Letter · Decision Letter 1]

17 Aug 2021

Gender with marital status, cultural differences, and vulnerability to hypertension : Findings from the national survey for noncommunicable disease risk factors and mental health using WHO STEPS in Bhutan

PONE-D-21-08522R1

Dear Dr. Imanaka,

We’re pleased to inform you that your manuscript has been judged scientifically suitable for publication and will be formally accepted for publication once it meets all outstanding technical requirements.

Kind regards,

Akihiro Nishi, M.D., Dr.P.H.

Academic Editor

PLOS ONE

Additional Editor Comments (optional):

Thank you so much for waiting for a while. The original two reviewer are now satisfied with the quality of the revised manuscript.

Reviewers' comments:

Reviewer's Responses to Questions

**Comments to the Author**

1. If the authors have adequately addressed your comments raised in a previous round of review and you feel that this manuscript is now acceptable for publication, you may indicate that here to bypass the “Comments to the Author” section, enter your conflict of interest statement in the “Confidential to Editor” section, and submit your "Accept" recommendation.

Reviewer #1: All comments have been addressed

Reviewer #2: All comments have been addressed

2. Is the manuscript technically sound, and do the data support the conclusions?

Reviewer #1: Yes

Reviewer #2: Yes

3. Has the statistical analysis been performed appropriately and rigorously? 

Reviewer #1: Yes

Reviewer #2: Yes

4. Have the authors made all data underlying the findings in their manuscript fully available?

Reviewer #1: Yes

Reviewer #2: Yes

5. Is the manuscript presented in an intelligible fashion and written in standard English?

Reviewer #1: Yes

Reviewer #2: Yes

6. Review Comments to the Author

Reviewer #1: The manuscript has been much improved and is in a nice condition now.

The author has fully responded to the reviewer's points.

Reviewer #2: (No Response)

7. PLOS authors have the option to publish the peer review history of their article (what does this mean?). If published, this will include your full peer review and any attached files.

Reviewer #1: No

Reviewer #2: No

---

## [Editor Report · Acceptance letter]

23 Aug 2021

PONE-D-21-08522R1 

Gender with marital status, cultural differences, and vulnerability to hypertension: Findings from the national survey for noncommunicable disease risk factors and mental health using WHO STEPS in Bhutan 

Dear Dr. Imanaka:

I'm pleased to inform you that your manuscript has been deemed suitable for publication in PLOS ONE. Congratulations! Your manuscript is now with our production department. 

Kind regards, 

on behalf of

Dr. Akihiro Nishi 

Academic Editor

PLOS ONE